# Using UAV Visible Images to Estimate the Soil Moisture of Steppe

**Fengshuai Lu** [1,2,†], **Yi Sun** [3,4,†] **and Fujiang Hou** [1,2,*]

1   State Key Laboratory of Grassland Agro-escosystems, College of Pastoral Agriculture Science and Technology, Lanzhou University, Lanzhou 730020, China; lufsh18@lzu.edu.cn
2   Key Laboratory of Grassland Livestock Industry Innovation, Ministry of Agriculture, Lanzhou University, Lanzhou 730020, China
3   Institute of Fragile Eco-environment, Nantong University, Nantong 226007, China; sunyi@ntu.edu.cn
4   School of Geographic Science, Nantong University, Nantong 226007, China
*   Correspondence: cyhoufj@lzu.edu.cn
†   Fengshuai Lu and Yi Sun contributed equally to this work.

**Abstract:** Although unmanned aerial vehicles (UAVs) have been utilized in many aspects of steppe management, they have not been commonly used to monitor the soil moisture of steppes. To explore the technology of detecting soil moisture by UAV in a typical steppe, we conducted a watered test in the Loess Plateau of China, quantitatively revealing the relationship between the surface soil moisture and the visible images captured using an UAV. The results showed that the surface soil moisture was significantly correlated with the brightness of UAV visible images, and the surface soil moisture could be estimated based on the brightness of the visible images of the UAV combined with vegetation coverage. This study addresses the problem of soil moisture measurement in flat regions of arid and semi-arid steppes at the mesoscale, and contributes to the popularization of the use of UAVs in steppe ecological research.

**Keywords:** grassland; grazing; vegetation coverage; visible image; brightness

## 1. Introduction

As the ecological defense of the earth, steppes expand the human land-use range [1]. Precipitation, temperature, and the relationship between them are the key determinants of the distribution, structure, and function of steppes globally [2]. The soil water of steppes mainly depends on precipitation at a large spatial scale and is affected both by human activities, including the grazing of livestock, irrigation, hay cutting, the improvement and cultivation of steppes, and precipitation at a regional or small spatial scale [3–5]. Grazing is one of the most important ways of steppe utilization, which has a certain effect on the soil moisture of steppes through a series of livestock disturbances [6]. In the short term, the intaking and trampling of livestock reduces vegetation coverage; increases the solar transmission area; makes the surface soil warm; and increases the evaporation of soil water, which further affects the growth of vegetation, including improving plant regrowth. The high-intensity tramping of livestock causes physical damage to the vegetation and soil; increases the compactness of soil; and decreases the permeability of air, which further prevents the infiltration of water from the surface to deep layers, and reduces the water and nutrient uptake by vegetation roots. The excessive excretion of livestock also affects the chemical elements of soil, such as organic carbon. Furthermore, the change in the soil chemical properties affects the distribution of soil water and vegetation [7]. Because of the key role of soil water in sustainable grassland management, a large number of methods have been developed to measure soil moisture, such as the conventional oven-dry method, time-efficient instrumental methods such as TDR (time domain reflectometry) [8,9], and modern remote sensing technology [10].

Remote sensing technology using unmanned aerial vehicles (UAVs) is a low-altitude remote sensing technique developed in recent years, with advantages such as high flexibility, high efficiency, and high resolution [11,12]. In addition, due to the ability of UAVs to fly under clouds, UAVs are being increasingly applied in steppe research [13]. The use of highly accurate aerial images to monitor vegetation coverage, biomass, soil patches, rat holes, the species diversity of vegetation, and livestock grazing is becoming a popular approach in the application of low-altitude remote sensing to ecology and biodiversity [14–21]. The UAV monitoring platform and decision tree algorithm could be used to construct an automatic tool to quickly, accurately, and automatically determine vegetation types at the landscape scale [22]. A model for estimating the FVC (Fractional Vegetation Cover) in the Gannan steppe based on large quadrat data obtained by a small UAV and an enhanced vegetation index had a high accuracy ($R^2 = 0.88$) [23]. UAV aerial images and a maximum entropy–genetic algorithm was used to estimate the vegetation coverage and biomass in the middle of the Hulunbuir steppe in Inner Mongolia and achieved good results [24]. The relative grazing intensity of a household pasture was evaluated by tracking and photographing the herd activity area by UAV [25]. UAV remote sensing was applied to the rapid, accurate, and efficient monitoring of the number of rat holes, which could greatly reduce the cost of manual monitoring and effectively monitor the occurrence of rat damage [26]. The images obtained from the specific UAV aerial photography patterns were used to generate point clouds so as to establish a canopy height model, which could be used to estimate the height of grassland vegetation [27]. The methods of remote sensing technology estimating the soil moisture in steppes include visible light–near infrared spectral reflection (e.g., the use of reflectivity and the Normalized Differential Vegetation Index (NDVI)) [28], thermal infrared spectral reflection (e.g., the use of thermal inertia and the Crop Water Stress Index (CWSI)) [29,30], fusion methods of visible light–near infrared and thermal spectral reflection (e.g., the use of the Temperature Vegetation Drought Index (TVDI) and Vegetation Water Supply Index (VSWI)) [31], and microwave remote sensing [32]. The images used in the above methods can be obtained by satellite, UAV, and thermal infrared instruments; however, the studies of using visible image to estimate soil moisture in steppes were rare. Putra et al. showed that the intensity index, the TGI index, and the ExGreen index of aerial visible images (red, green, and blue bands) could be carried out to determine trends of soil moisture in agricultural land, where the TGI index has a higher coefficient of determination [33]. Zanett et al. used RGB images and artificial neural networks to estimate soil moisture [34]. Dos Santos et al. established different linear models using RGB, HSV (Hue, Saturation, Value), and the digital number of a panchromatic image in each type of soil [35]. However, most of these studies were controlled experiments and conducted without vegetation interference. In this study, the purpose is to estimate the soil moisture in steppes using aerial visible images under natural conditions. Previous studies have noted that the topsoil reflectivity was closely related to the soil moisture [36]. Liu et al. established an empirical model of soil moisture using the data of measured soil moisture and topsoil reflection [37]. Kolassa et al. identified that the synergy between the brightness temperature data of AMSR-E (a microwave radiometer on satellite) and the backscatter data of ASCAT (a microwave scatterometer on satellite) could effectively improve the quality of estimation of soil moisture [38]. Zhu et al. indicated that diffuse reflectance spectroscopy was a great way to estimate surface soil moisture by utilizing the topsoil reflectivity to quantify the soil moisture [39].

In the steppe ecosystem, precipitation has a direct effect on the surface soil water; soil water is the main factor affecting the formation and evolution of vegetation patterns [40,41], and the conditions of soil and vegetation under fluctuating precipitation could be reflected by aerial visible images (Figure 1). The measurement of the soil moisture of steppes is a routine investigation of the steppe ecosystem; it is an important way to extract the surface parameterization of steppes and assess changes in climate. In this study, we assume that the characteristic factor of UAV visible images could estimate the surface soil moisture such as topsoil reflectivity. We would obtain UAV visible images with different gradients of soil moisture in a typical steppe; seek the characteristic factor by digital image processing; further establish estimation models of surface soil moisture by considering the characteristic factor, soil moisture, vegetation, and other factors; and finally verify the estimation models using the data of grazing land.

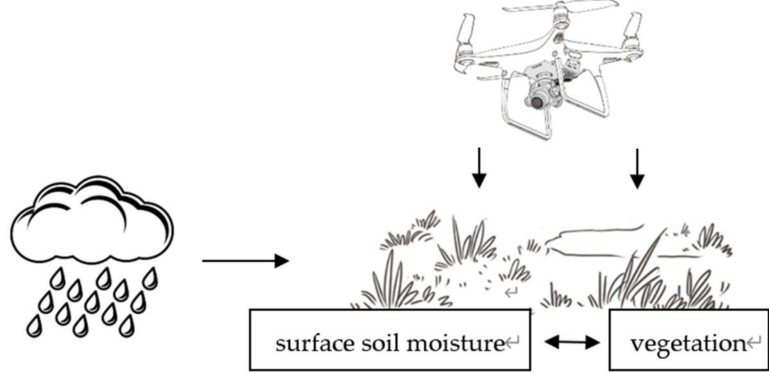

**Figure 1.** Schematic diagram of research.

## 2. Materials and Methods

### 2.1. Study Site

The study site is located in the Huanxian Grassland Agriculture Trial Station of Lanzhou University, Tianshui Township, Huanxian County, Gansu province, China (37°3′ N, 109°38′ E, a.s.l. 1650 m). The climate is cataloged as a Temperate Continental Monsoon Climate. The annual mean temperature is 7.5 °C, and the annual mean precipitation is about 350 mm, over 70% of which takes place between late June and September. The annual mean evaporation potential is about 1675 mm, and the annual mean sunshine duration is 2596.1 h [42]. The study site is classified as a typical steppe in terms of the Grassland Comprehensive Order Classification [43]. The plant growth period is from late March to early September. The dominant species include *Stipa bungeana*, *Artemisia capillaris*, and *Lespedeza bicolor*. The type of agricultural system is characterized by the integrated crop/grassland-livestock production system [44].

### 2.2. Study Design

Since 2001, a long-term grazing experiment has been performed with free grazing and rotational grazing [45]. In the free grazing sites, we conducted the watered test in 2019 to explore the estimation of soil moisture by UAV. Meanwhile, in the rotational grazing sites, we set 48 fixed quadrats to verify the practicability of the estimation.

The reasons for the watered test in the study site were: (1) this area was located at the convergence zone of the Loess Plateau and the Maowusu Desert, and the average annual evaporation is about 6 times the precipitation; (2) the soil surface is covered with a lot of loess with a loose texture, and it is easy for the surface water to infiltrate to deep layers; (3) the vegetation over the ground is relatively low and sparse, and there is an obvious distribution of bare land [45,46]. Due to the above reasons, the stable value of the 0–10 cm soil moisture in this area is generally below 5%, with limited variances. Therefore, in order to obtain different gradients of soil moisture, it is necessary to simulate precipitation.

### 2.3. Sample Setting and Measurement of Soil Moisture

The rotational grazing sites have been used from early June to mid-September every year, and the grazing period was from 8:30 to 18:30 every day, 10 days per month. There are four stocking rates with three replicates for each, and each paddock is 50 m × 100 m [45]. Twenty-five local Tan sheep were used for grazing experiment. The stocking rates were calculated by the following formula:

$$SR = \frac{LN}{PA}. \tag{1}$$

In the formula, *SR* is the stocking rate, LN is the livestock number of each stocking rate, and PA is the total paddock area of each stocking rate. The stocking rate of the rotational grazing land was

0 (no grazing), 2.67, 5.33, and 8.67 sheep/ha, respectively. Outside the rotational grazing sites are the free grazing sites.

Ten 2 m × 2 m quadrats were set up in the free grazing sites with flat terrain and representative, consistent vegetation, of which five were watered and the remaining five were used as controls. The watered quadrats were fixed, while the comparative quadrats were not fixed and treated. Three repeating watered tests were conducted in early June, early July, and early August during the vegetation growth period. For the watered quadrats, a waterproof plastic sheet was inserted vertically to underground (0–50 cm) around the quadrats to isolate them from the surrounding soil, and meanwhile we used PVC board with a height of 5 cm (above the ground) to isolate the watered quadrats from the surrounding ground to prevent the inflow of surface water. In the case of rain during the experiment, a 3 m × 3 m transparent waterproof plastic sheet was used keep out the rain, and the sheet was removed after the rain. In each watered test, water was poured at the same speed to the surface of the quadrat with a sprinkling pot by hand. The standard watering amount was for a saturated soil moisture of 0–10 cm, where the saturated soil moisture refers to the maximum water content of the soil [36]. In each test, the watering amount was measured as 500–550 mL per quadrat, with a saturated soil moisture at 0–10 cm of 25–30%. The test period was from the day of watering to the day when the 0–10 cm soil moisture was less than 5%, because the stability value of 0–10 cm soil moisture was usually less than 5% in the study site. During the period of each watered test, the soil moisture of the 0–10 cm layer was measured by a measuring instrument (TZS-ECW-G) every noon, and each quadrat was measured four times, then we calculated the average value. The TZS-ECW-G (Zhejiang Top Instrument Co., Ltd., Hangzhou, China) measures the volumetric soil moisture with a resolution of 1%; the range of measurement is 0–100%, the response time is 2 s, and the relative percentage error is less than 3%. When we use it to collect the value of soil moisture, its water sensor needs to be inserted into the soil firstly.

In the rotational grazing land, four 2 m × 2 m quadrats were set up in each paddock. A total of 48 2 m × 2 m quadrats were taken as the verification quadrats. The soil moisture of the verification quadrats was measured when the 0–10 cm soil moisture was stable and after rainfall, respectively. The measurement method was the same as that above—that is, the soil moisture of 0–10 cm was measured by TZS-ECW-G every day at the local noon, and each quadrat was measured four times, then we calculated the average value.

### 2.4. Processing of UAV Visible Images

Because the UAV visible images were RGB images composed of soil and vegetation, the concept of the "average pixel brightness value" (cd/m$^2$) (hereinafter, referred to as "brightness") was introduced. The value was the pixel value of the RGB image defined by a computer, which reflected the brightness difference of the RGB images. It was calculated from the formula: Y (brightness) = 0.299 × R + 0.587 × G + 0.114 × B, where R, G, and B represent the value in the red, green, and blue channels of the RGB image, respectively [47]. The brightness of an object had two determining factors, which were the intensity of the light received by the object and the object's ability to reflect the light. Water was an important factor affecting the reflection ability of vegetation and soil to light. The typical steppe was a representative area with sparse vegetation and an exposed surface; therefore, the brightness was considered as the research index of UAV visible images.

The precautions in the acquisition phase of the UAV visible images were as follows: (1) the brightness of the UAV visible images was affected by the light intensity, flight speed of the UAV, and the visible camera parameters (model, lens angle, ISO, aperture, and shutter speed). Among these influencing factors, the flight speed of the UAV and visible camera parameters (model, lens angle, ISO, aperture, and shutter speed) were controllable, so appropriate UAV parameters were set and no change was made during the whole test. As an uncontrollable influence factor, the light intensity was affected by the solar altitude angle and the atmospheric transparency. In order to unify the solar altitude angle, the UAV visible images were captured from 12:00 to 13:00 (local noon time). The errors caused by the

atmospheric transparency would be corrected in the later period. (2) to match the soil moisture in time and space, the UAV visible images were taken synchronously with the measurement of the soil moisture of the 0–10, 10–20, and 20–30 cm layers. In the case of rainy weather at noon, the taking of the UAV visible images would be postponed accordingly.

In the free grazing sites, a DJI Phantom 3 Professional (DJI-Innovations, Shenzhen, China) was used daily to take visible images for the watered and comparative quadrats. The DJI Phantom 3 Pro is equipped with a 4 K camera lens, which can take 12-megapixel static photos. The camera parameters of the DJI Phantom 3 Pro were set as shown in Table 1. The camera parameters were not changed in the watered test. Every UAV visible image must cover 2 m × 2 m quadrat area containing all the information, therefore we set the coverage area be 3 m × 3 m, and then calculated that the height was 1.4 m according to the lens angel of the camera and the coverage area (Figure 2). Combining the above aspects, the UAV flight mode was a speed of 3 m/s, a height of 1.4 m, vertical shooting on every point, and a coverage area of 3 m × 3 m for each image. In the rotational grazing land, we used the same flight mode by the DJI Phantom 3 Pro to take visible images for verification quadrats.

**Table 1.** The camera parameters of the DJI Phantom3 Professional.

| Parameter | Value | Parameter | Value |
|---|---|---|---|
| image sensor | 1-inch CMOS | aperture | f/2.8 |
| lens angel | FOV 94° 20 mm | ISO | 200 |
| photo size | 4864 × 3648 pixels | shutter speed | 1/320 s |

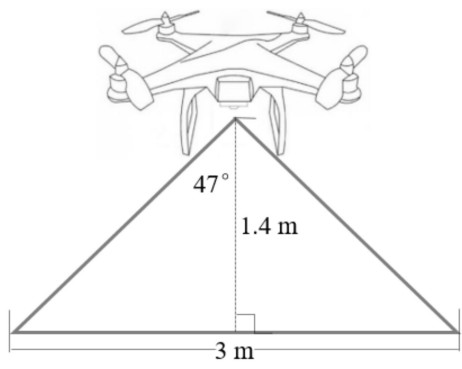

**Figure 2.** Schematic diagram of the flight height.

In order to correct each individual image of the test, the "Mosaic" flight mode was also used to acquire overlapping aerial photographs and generate a georeferenced orthomosaic of the study area (Figure 3) based on FragMAP [48]. Briefly, corners 1–4 of the study area were set manually on an iPad (carrier of FragMAP, Huawei M5, Shenzhen, China), then way points were automatically determined based on the specified fly height, speed, and interval of the shoot and overlap rate (height was set at 20 m in this study, and resolution was <1 cm). During each flight, the UAV took photographs at intervals of 3 s automatically. We generated a georeferenced orthomosaic of the study area with Pix4Dmapper (Pix4D S.A., Lausanne, Switzerland) (Figure 3). Aerial photographs of the quadrat were trimmed to the minimal size required to cover the study area and control points. They were ortho-rectified according to the georeferenced orthomosaic, using at least 10 reference points in ArcGIS (10.2.2). To find enough control points, any fixed object was regarded as a control point—e.g., a grasscluster, a big stone, etc. [49].

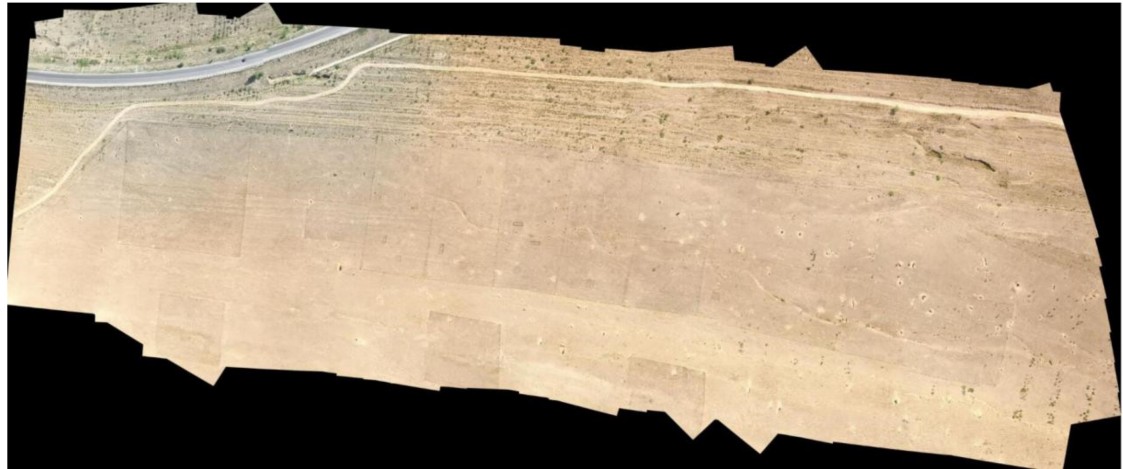

**Figure 3.** Georeferenced orthomosaic of the study area.

### 2.5. Control of Influencing Factors and Correction of Errors

As a research index of UAV visible images, the brightness depends on dual changes in the vegetation and soil, and is affected by a series of influencing factors. Among these factors, controllable factors have been controlled in the acquisition phase of UAV visible images. The uncontrollable factor is the light intensity, and the atmospheric transparency is the main factor affecting the light intensity. The reason is that visible light-electromagnetic radiation is transmitted in the sun-surface-sensor (UAV) route, and the transmission process is affected by the absorption and scattering of atmospheric molecules, water vapor, aerosol, and other atmospheric components. The visible light-electromagnetic radiation finally accepted by UAV not only includes the surface reflection information, but also records the interference of the atmosphere in the surface reflection information [50].

To correct the error caused by atmospheric transparency among different test dates, the UAV visible images were taken on sunny dates with less than 10% cloud cover according to meteorological information and actual observation. On this basis, we had selected a comparative quadrat as a standard quadrat. As an only uncontrollable influence factor, the change in the brightness of the standard quadrat with the change in the atmospheric transparency was fixed, therefore we increased or decreased the brightness of the watered quadrats and comparative quadrats according to the standard quadrat.

### 2.6. Data Analysis

SPSS 20.0 was used to conduct the statistical analysis—i.e., a goodness-of-fit test (Shapiro–Wilk test, univariate procedure) was used to test the normality of the data. A Poisson correlation analysis was applied to test the correlation among the soil moisture, brightness, and vegetation coverage. The coefficient of determination ($R^2$) and its $p$ values were used to evaluate the regression between the soil moisture and brightness and the vegetation coverage. A one-way ANOVA was applied to analyze the soil moisture among stocking rates. The vegetation coverage was extracted using ImageJ. ImageJ was also used to draw histograms, grayscale images, and 3D Surface Plots of the UAV visible images.

#### 2.6.1. Data Analysis of Brightness

An UAV visible image containing regions with different 0–10 cm soil moistures; the 0–10 cm soil moisture was 22.1%, 14.1%, and 4.6% from the left to right in June (Figure 4a). The ImageJ software was used to output the brightness value of the UAV visible images.

(a)　The UAV visible image was cropped into three parts according to the different soil moistures (Figure 4b), then we put them into ImageJ for processing. Firstly, the visible images were converted to 8-bit grayscale images. A grayscale image has 256 levels of brightness, where 0 represents the darkest and 255 represents the brightest. To significantly layer the brightness of

the grayscale images, a spectral filter was added to the grayscale images, and thereby we divided the 256 levels of brightness into seven colors roughly. The definition from dark to bright was red, orange, yellow, green, cyan, blue, and purple. The brightness gradually increased from left to right (Figure 4b). The brightness of the vegetation is lower than that of the soil, and the lower the 0–10 cm soil moisture was, the more obvious the brightness contrast was.

(b)  To intuitively compare the differences in brightness, 3D Surface Plot was used to convert the brightness into height information, transforming the concept of dark to bright into deep to shallow (Figure 4c).

(c)  Finally, the histograms were output. In these histograms, "Mean" refers to the brightness that we talked about. According to Figure 4d, 22.1%, 14.1%, and 4.6% of the 0–10 cm soil moisture corresponded to brightnesses of 100.57, 117.05, and 130.628 cd/m$^2$, respectively. The differences in soil moisture at 0–10 cm were reflected in the brightness of the UAV visible images.

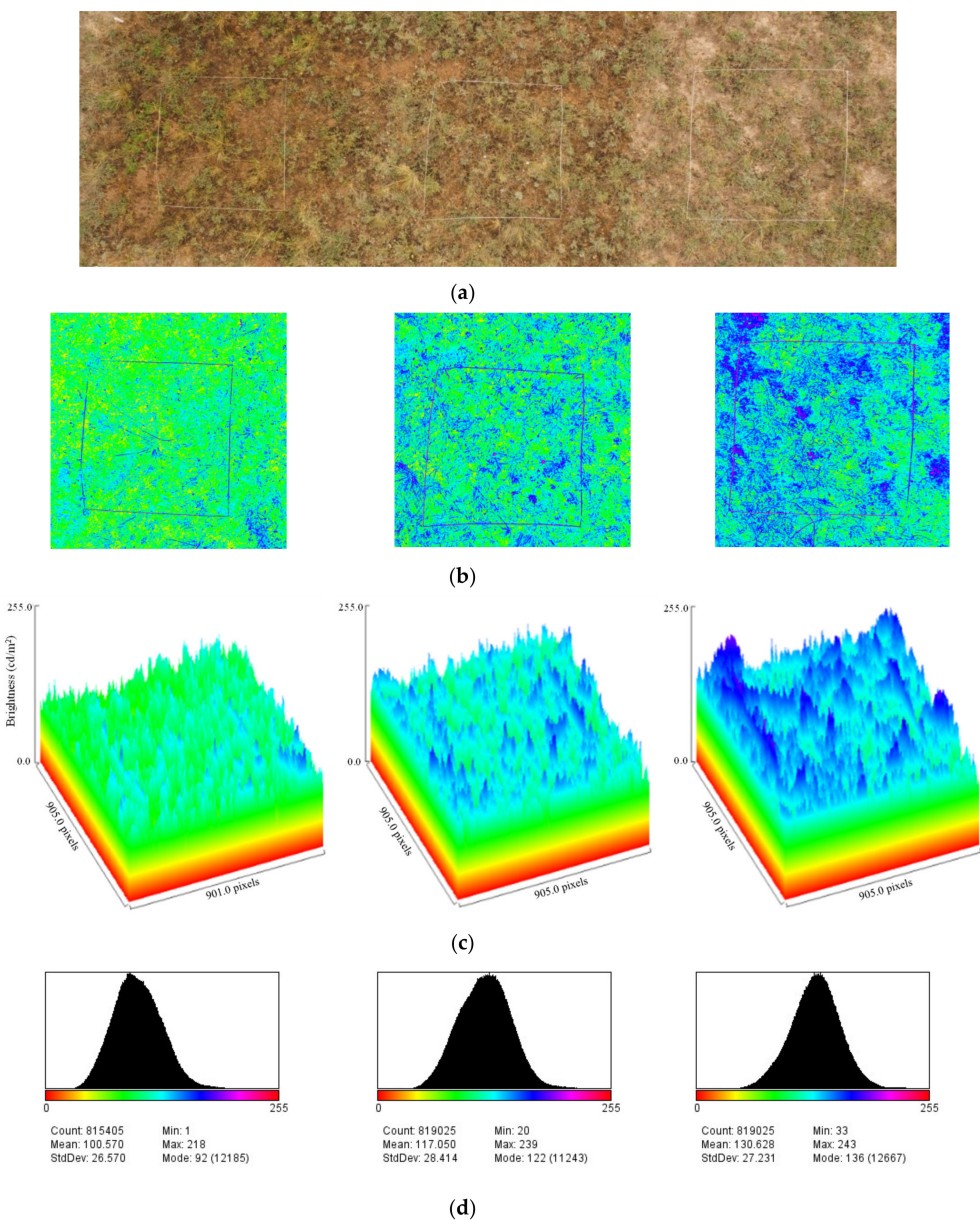

**Figure 4.** Processing of aerial visible images; (**a**) UAV visible image, (**b**) Eight-bit grayscale images with spectral filter, (**c**) 3D surface plot, (**d**) Histogram.

### 2.6.2. Data Analysis of Vegetation Coverage

ImageJ could divide an UAV visible image into vegetation coverage and soil coverage on the basis of the appropriate HSB (hue, saturation, and brightness) settings. The extraction process is as follows: (a) open the image in ImageJ, then click the option of threshold color, and adjust the HSB to the optimal values to select the vegetation coverage portion. In this study, the HSB setting was as shown in Figure 5. (b) Click the mask function to adjust the vegetation coverage portion. (c) Preserve the final selected portion and output the result of the vegetation coverage area.

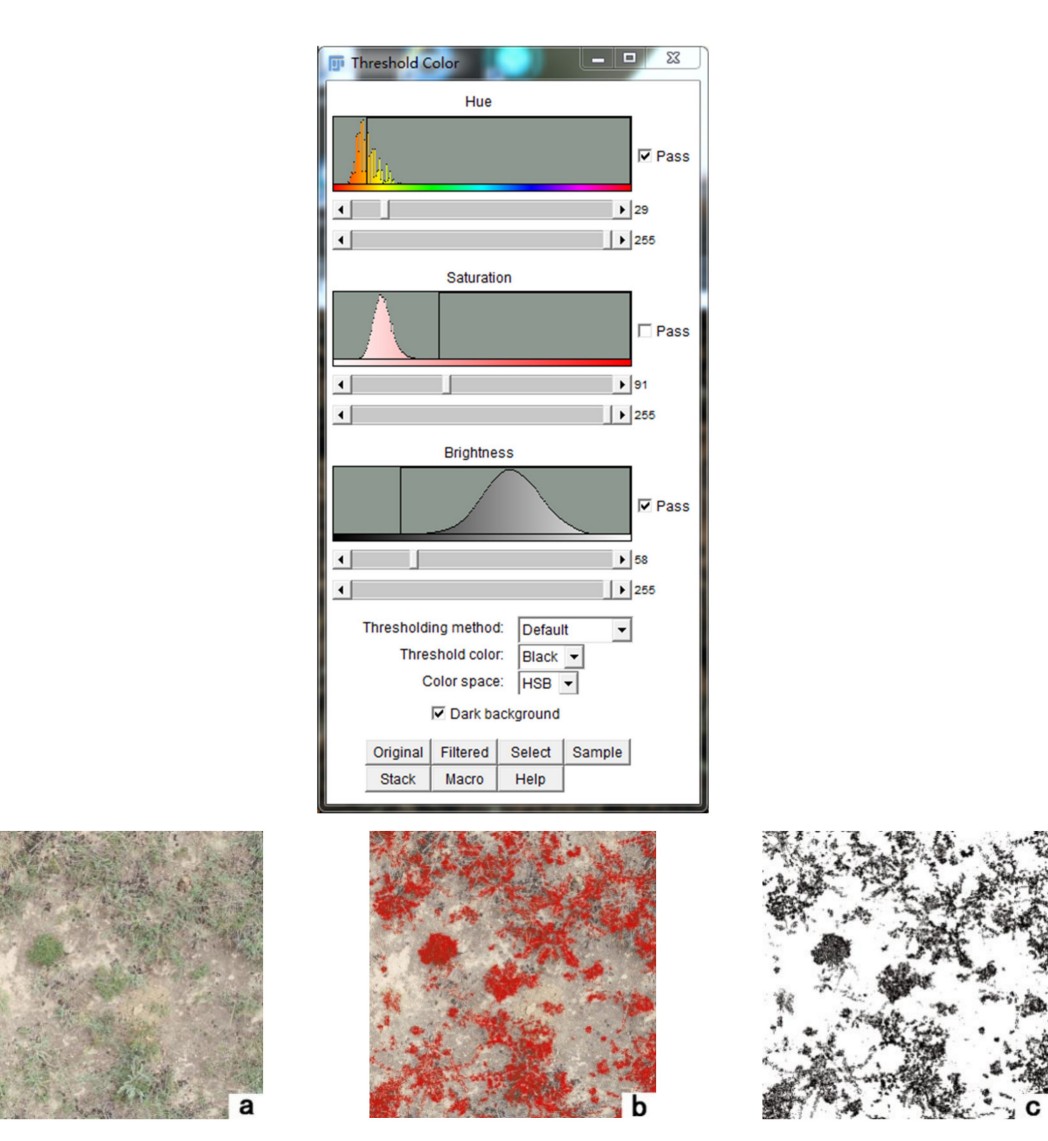

**Figure 5.** The HSB (hue, saturation, and brightness) setting of obtaining vegetation coverage. (**a**) Original UAV visible image; (**b**) final selected portion; (**c**) result of the vegetation coverage area.

### 2.7. Data Analysis of Verification Quadrats

In order to verify the practicability of the estimation models of the 0–10 cm soil moisture, we had analyzed the fitting effect of the measured value and the estimated value using verification quadrats. The absolute values of the total relative error (RS) and average relative error (RMA) were used for the verification. In general, when the absolute value of RS was less than 20% and the absolute value of RMA

was less than 20%, the estimation model would meet the requirement of practicability. The formulas are as follows:

$$RS = \left( \frac{\sum_{i=1}^{N} Yi - \sum_{i=1}^{N} yi}{\sum_{i=1}^{N} yi} \right) \times 100\%, \tag{2}$$

$$RMA = \frac{1}{N} \left( \sum_{i=1}^{N} \frac{|Yi - yi|}{|yi|} \right) \times 100\%, \tag{3}$$

where Y represents the measured value, y represents the estimated value, and N represents the number of verification quadrats.

## 3. Results

### 3.1. Estimation of 0–10 cm Soil Moisture by UAV

The 3D surface plots of the UAV visible images of the watered quadrats were generated. The purpose was to compare the early stage and the late stage of watering. The watered quadrats were darker in the early stage and brighter in the later stage in early June, early July, and early August (Figure 6).

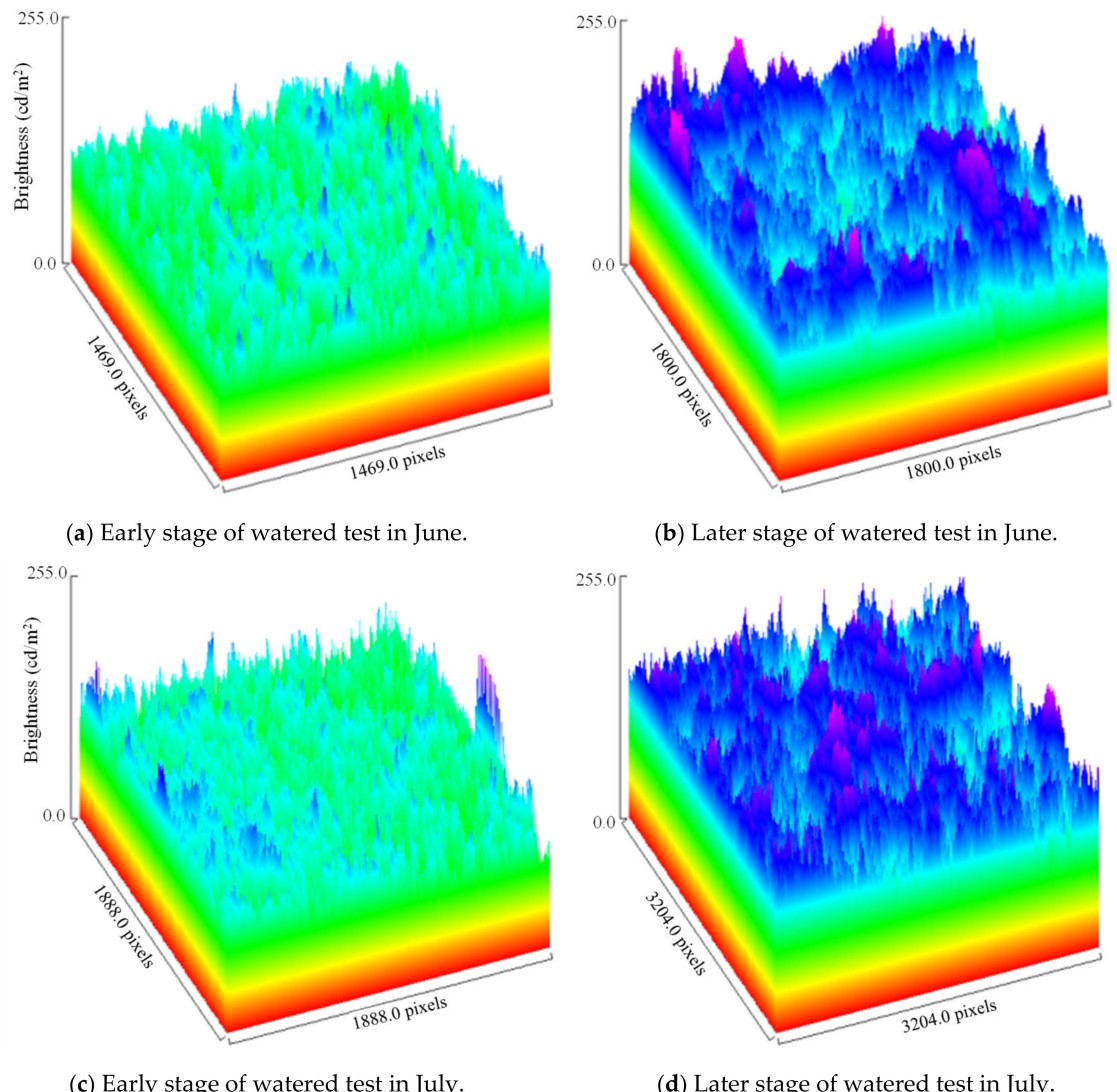

(**a**) Early stage of watered test in June.   (**b**) Later stage of watered test in June.

(**c**) Early stage of watered test in July.   (**d**) Later stage of watered test in July.

**Figure 6.** *Cont.*

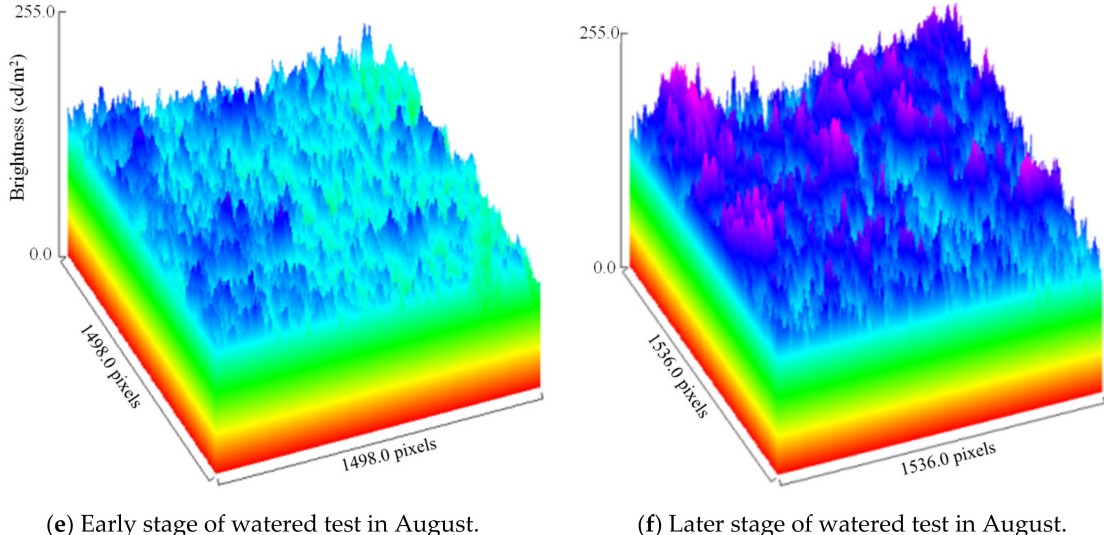

(**e**) Early stage of watered test in August.　　(**f**) Later stage of watered test in August.

**Figure 6.** 3D surface plot of the visible images of watered quadrats.

The timeline chart of the 0–10 cm soil moisture and the brightness of watered quadrats was established to analyze the specific process of the change of brightness (Figure 7). The 0–10 cm soil moisture decreased gradually with the observation day, and the brightness of the UAV visible images increased gradually with the observation date. It took 7, 6, and 5 days for the 0–10 cm soil moisture to reach stable values of 2.73%, 3.41%, and 4.17% in June, July, and August, respectively. According to the local meteorological station, the temperature was highest in early August, indicating that the 0–10 cm soil moisture was greatly affected by temperature. The maximum value of brightness appeared in June, and the minimum value of brightness appeared in August; the reason for this was that vegetation coverage had an impact on the brightness value. Then, we extracted the vegetation coverage of each UAV visible image.

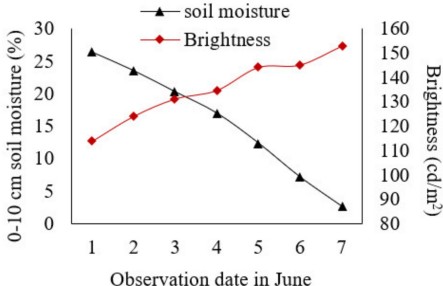

(**a**) The 0–10 cm soil moisture and brightness in June

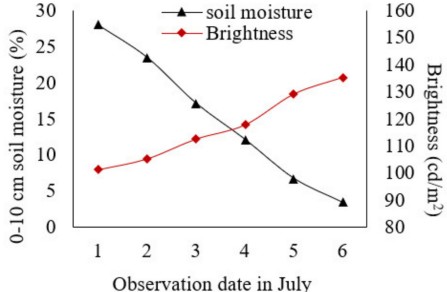

(**b**) The 0–10 cm soil moisture and brightness in July

**Figure 7.** *Cont.*

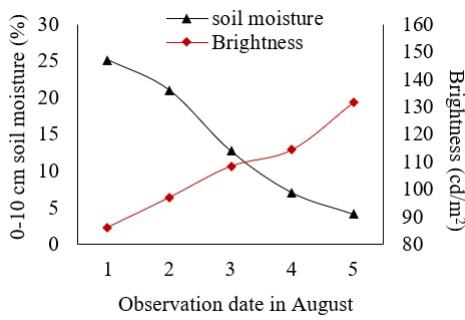

(c) The 0–10 cm soil moisture and brightness in August

**Figure 7.** The 0–10 cm soil moisture and brightness.

There was a significant relationship between the 0–10 cm soil moisture and the brightness, this is because generally the required value of correlation index is large than 0.3, in this study, the correlation indexes were large than 0.3 in different situations (Table 2). In addition, there was a significant relationship among the 0–10 cm soil moisture, the brightness and the vegetation coverage when the 0–10 cm soil moisture was stable (Table 2). Therefore, the estimation of the 0–10 cm soil moisture was divided into two situations: (1) the 0–10 cm soil moisture was stable; (2) the 0–10 cm soil moisture was larger than the stable value. A correlation analysis was conducted on the factors of the estimation of the 0–10 cm soil moisture (Table 2). When the 0–10 cm soil moisture was larger than the stable value, the 0–10 cm soil moisture had a significant negative correlation with the brightness ($p < 0.001$). When the 0–10 cm soil moisture was stable, the 0–10 cm soil moisture had a significant negative correlation with the brightness ($p < 0.001$) and a significant positive correlation with the vegetation coverage ($p < 0.001$).

**Table 2.** Correlation analysis of factors affecting the soil moisture estimation.

| Index | 0–10 cm Soil Moisture | Brightness | Vegetation Coverage |
| --- | --- | --- | --- |
| 0–10 cm soil moisture | - | −0.6 *** | 0.434 *** |
| Brightness | −0.787 *** | - | 0.232 * |
| Vegetation coverage | −0.024 | 0.092 | - |

Note: The lower left part is the analysis when the 0–10 cm soil moisture was larger than the stable value, and the upper right part is the analysis when the 0–10 cm soil moisture is stable. *, **, and *** mean significant at the levels of 0.05, 0.01, and 0.001, respectively.

The 198 visible images were taken by UAV to quantitatively analyze the effect of the 0–10 cm soil moisture and vegetation coverage on brightness, and established the estimation models of the 0–10 cm soil moisture using linear regression and multiple regression methods. When the 0–10 cm soil moisture was larger than the stable value, the determination coefficient of predicting the 0–10 cm soil moisture based on the brightness was significant at the 0.001 level, indicating that the 0–10 cm soil moisture could be predicted by a linear regression model between the 0–10 cm soil moisture and brightness. When the 0–10 cm soil moisture was stable, the determination coefficient for predicting the 0–10 cm soil moisture based on the brightness and vegetation coverage was significant at the 0.001 level, indicating that the 0–10 cm soil moisture could be predicted by a multiple regression model of the 0–10 cm soil moisture, brightness, and vegetation coverage (Table 3).

**Table 3.** Estimation models of the 0–10 cm soil moisture.

| Application Condition | Prediction Model | $R^2$ | Sig |
|---|---|---|---|
| 0–10 cm soil moisture > stable value | $Y = 70.52 - 0.47X_1$ | 0.77 | <0.001 |
| 0–10 cm soil moisture = stable value | $Y = 23.39 - 0.2X_1 + 0.07X_2$ | 0.86 | <0.001 |

Note: "Y" represents the 0–10 cm soil moisture; "$X_1$" represents the brightness; "$X_2$" represents the vegetation coverage.

### 3.2. Practical Verification of Estimation Models of 0–10 cm Soil Moisture

In the rotational grazing sites, the livestock intake reduced the coverage of edible vegetation, made the ground temperature rise, and increased the evaporation of soil water. On the other hand, sheep trampling increased the compactness of soil and then reduced the infiltration of soil water. When the 0–10 cm soil moisture was at a stable value, the 0–10 soil moisture was significantly different among these plots with four stocking rates (Table 4). As a whole, the 0–10 cm soil moisture in the 0 and 2.67 sheep/ha plots was significantly higher than that in the 5.33 and 8.67 sheep/ha plots ($p < 0.05$), and there was no significant difference between the 0 and 2.67 sheep/ha plots. Therefore, the 96 verification quadrats in the rotational grazing sites could verify the estimation model. When the 0–10 cm soil moisture was larger than the stable value, the soil moisture after rainfall could be used as a verification quadrat. The comparison of the 0–10 cm soil moisture between the measured value and the estimated value, where the determination coefficient and the values of RS and RMA met the verification requirements, indicated the estimation models of the 0–10 cm soil moisture had a certain practicality (Figure 8).

**Table 4.** Soil moisture (0–10 cm) under four stocking rates.

|  | 0 Sheep/Ha | 2.67 Sheep/Ha | 5.33 Sheep/Ha | 8.67 Sheep/Ha |
|---|---|---|---|---|
| June | 5.66 ± 0.08 a | 5.41 ± 0.13 a | 4.23 ± 0.1 b | 3.7 ± 0.4 c |
| July | 6.68 ± 0.63 a | 5.81 ± 0.42 a | 4.64 ± 0.4 b | 4.52 ± 0.48 b |
| August | 6.29 ± 1.01 a | 5.45 ± 0.23 a | 3.74 ± 0.92 b | 3.08 ± 0.14 b |

Note: Different lowercase letters (i.e., a, b, and c) indicate significant differences in the soil moisture (0–10 cm) under different stocking rates in each month at the level of $p < 0.05$ (based on a statistical analysis of a one-way ANOVA).

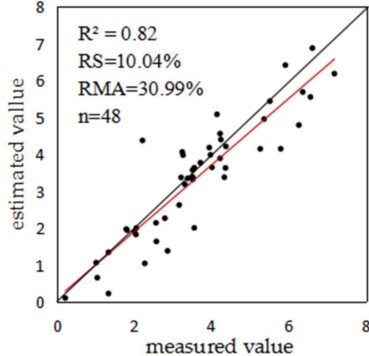

(**a**) When the soil moisture was at a stable value.

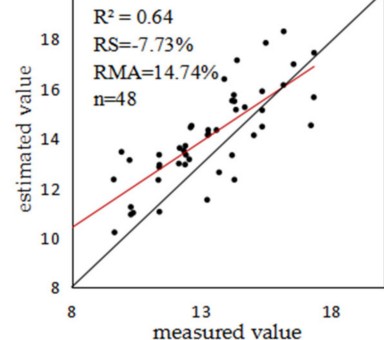

(**b**) When the soil moisture was larger than the stable value.

**Figure 8.** The comparison of the 0–10 cm soil moisture between the measured value and the estimated value.

## 4. Discussion

In the field of processing satellite remote sensing images, Digital Number (DN) refers to the pixel brightness value of images, and each pixel in the image had its corresponding DN. The appearance reflectivity, surface reflectivity, and surface albedo are all obtained from the DN in images. In the field

of processing UAV visible images, the "average pixel brightness value" is defined by a computer to reflect the brightness of images; it is generally used for the adjustment of image color and correction of photographic exposures. Surface reflectivity was usually well correlated with soil moisture [51]. Surface reflectivity refers to the ratio between the surface reflection of solar radiation and the total solar radiation. Each band had a corresponding surface reflectivity [52]. Generally, the reflectivity of a rough surface was smaller than that of a smooth surface [53], the reflectivity of a land surface was smaller than that of a snow surface [54], and the reflectivity of a moist surface was smaller than that of a dry surface [55]. By this study, we confirmed that the "average pixel brightness value" of UAV visible images could estimate the surface soil moisture like topsoil reflectivity.

As one of the ways of solar energy transmitting, visible light-electromagnetic radiation is emitted from the sun and reaches the earth's surface through various atmospheric components. In the transmitting process, part of the visible light-electromagnetic radiation was absorbed and scattered by the atmosphere, part is absorbed by the surface, and the rest is reflected by the surface [56]. In this study, due to the close distance from the surface to the UAV, the effects of the atmosphere on this route were negligible, but the effects of the atmosphere on the sun-surface route still exists (Figure 9); therefore, the correction of the observed result was necessary.

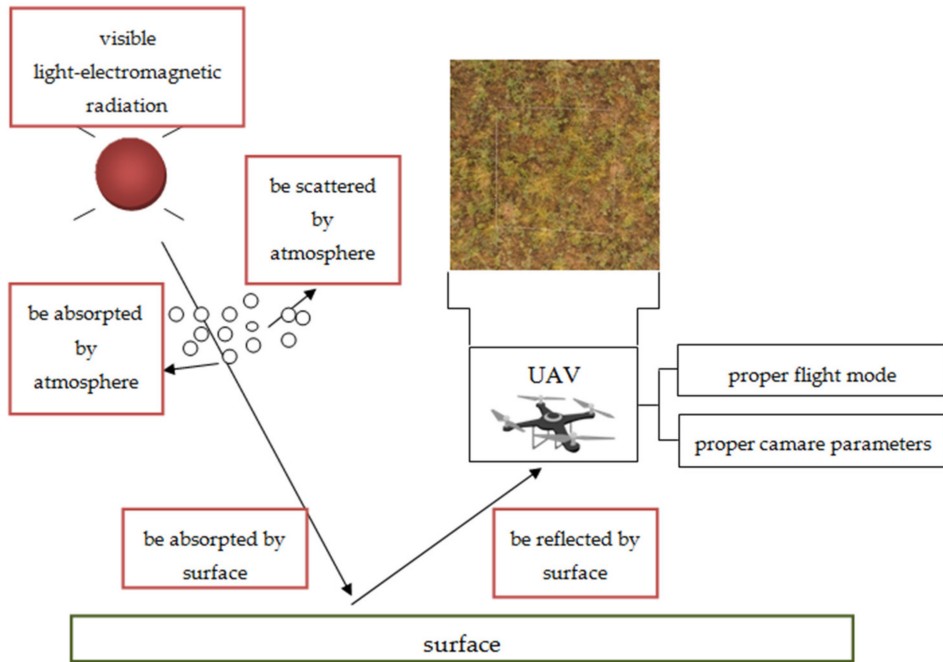

**Figure 9.** The conceptual model of UAV monitoring the soil moisture.

We took the average pixel brightness value as the characteristic factor of UAV visible images, and found that it could better indicate differences in the 0–10 cm soil moisture of a typical steppe in the Loess Plateau. When water is immersed in the soil, the sun's rays have to first pass through the water to reach the surface soil. Similarly, the reflected rays had to first pass through the water to reach the camera of the UAV, thus there was a loss of the reflected rays, as well as in the refraction and absorption process [57,58].

When the 0–10 cm soil moisture was larger than the stable value, the 0–10 cm soil moisture had a significant negative correlation with the brightness. When the 0–10 cm soil moisture was stable, the 0–10 cm soil moisture had a significant negative correlation with the brightness and a significant positive correlation with the vegetation coverage (Table 2). The surface reflection of the visible light-electromagnetic radiation of humid soil is less than that in dry soil, and the surface reflection of vegetation also decreases in humid soil environments; therefore, vegetation coverage and soil moisture have the same influences on the brightness. The surface reflection of the visible

light-electromagnetic radiation of dry soil was relatively larger, while that of vegetation was relatively small; therefore, vegetation coverage and soil moisture had inverse influences on the brightness [55].

The traditional oven-dry method and time-efficient instrumental method are suitable for a point scale to measure soil moisture and have obvious limitations for large- or medium-scale use. The majority of modern remote sensing technology concentrates on non-single visible bands; most of them rely on multiband operations to measure soil moisture [29], such as infrared camera drones and high-cost and complex digital image processing [31]. The estimation technology using UAV could solve these problems. Firstly, the resolution of an aerial remote sensing image is higher than that of satellite remote sensing image, so its interpretation and classification accuracy is higher. Secondly, aerial remote sensing images can be interpreted in stereo, while satellite remote sensing images can only be interpreted in plane. Thirdly, being compared with aerial remote sensing images, satellite remote sensing images have a larger scale and are more difficult to interpret than aerial remote sensing images. In this study, we adopted reverse water regulation on a typical steppe, simulated different evaporation stages after precipitation, observed the characteristics of aerial images of vegetation and soil water at different stages, and then tested estimation models by utilizing grazing lands. The results showed that this method was feasible. In addition, the vegetation coverage index was added into the estimation model, expanding the geographical region to estimate the soil moisture. It is a new mode to estimate the soil moisture in steppes using UAV; if the conditions are right, the visible image processing technique could be used for other steppes as well.

## 5. Conclusions

In the typical steppe of the Loess Plateau, the average pixel brightness value of UAV visible images could be used as a factor to estimate the 0–10 cm soil moisture. Two estimation models of the 0–10 cm soil moisture were established. When the 0–10 cm soil moisture was at a stable value, the model was a multiple regression model among the soil moisture (Y), the average pixel brightness value ($X_1$), and the vegetation coverage ($X_2$): $Y = 23.387 − 0.179X_1 + 0.073X_2$ ($p < 0.001$). When the 0–10 cm soil moisture was larger than the stable value, the model was a general linear model between the soil moisture (Y) and the average pixel brightness value ($X_1$): $Y = 70.519 − 0.472X_1$ ($p < 0.001$). The soil moistures of the 0–10, 10–20, and 20–30 cm layers were positively correlated ($p < 0.001$).

When the 0–10 cm soil moisture was at a stable value, and the determination coefficient ($R^2$) of the estimation model, the absolute values of the RS and RMA, and the coefficient ($R^2$) between the measured value and the estimated value were 0.86, 10.04%, 30.99%, and 0.82, they all met the practical requirements. When the 0–10 cm soil moisture was larger than the stable value, and the determination coefficient ($R^2$) of the estimation model, the absolute values of the RS and RMA, and the coefficient ($R^2$) between the measured value and the estimated value were 0.77, 7.73%, 14.74%, and 0.64, they all met the practical requirements.

On the premise of controlling the influencing factors and correcting the test errors, it was feasible to evaluate the 0–10 cm soil moisture by UAV visible technology in a typical steppe. It provided practical significance for the evaluation of soil moisture in arid and semi-arid steppe. In addition, it contributes to the popularization of UAVs in steppe ecological research.

**Author Contributions:** Conceptualization and supervision, F.H.; Design and methodology, F.H. and F.L.; field investigation, F.L.; software, F.L.; validation, F.H. and Y.S.; data analysis, F.H. and F.L.; data curation, F.H. and F.L.; writing—original draft preparation, F.H. and F.L.; writing—review and editing, F.H. and Y.S. All authors have read and agreed to the published version of the manuscript.

**Funding:** National Natural Science Foundation of China (No. 31672472), Program for Changjiang Scholars and Innovative Research Team in University (IRT_17R50), and Key R & D Program of Ningxia Hui Autonomous Region (2019BBF02001).

**Conflicts of Interest:** The authors declare no conflict of interest.

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
