# Peer review of "Using UAV Visible Images to Estimate the Soil Moisture of Steppe"

_water, doi:10.3390/w12092334_

Round 1
Reviewer 1 Report
The manuscript describes the results from the images acquired by a drone of a steppe in China in relation to the ability to extract the soil moisture content through the images. In general the manuscipt is well written and in good language.
Some specific points:
- The title doesnt reflect the impact of the work. Although the results are only on the Loess Plateau in China, if the conditions are right, the image processing technique could be used for other steppe as well.
- Table 4 doesnt look right. Line numbers are embedded in the table for some reason.
- The novelty of the work is not explained at all. Compared to other available technologies, what is the benefit of using this technology?
Author Response
Reviewer #1's comments:
Point 1: The title does not reflect the impact of the work. Although the results are only on the Loess Plateau in China, if the conditions are right, the image processing technique could be used for other steppes as well.
Response 1: Thanks for your comment. We have changed the according to your suggestion.
“Using UAV visible image to estimate soil moisture of steppe” (Line 2-3)
Point 2: Table 4 does not look right. Line numbers are embedded in the table for some reason.
Response 2: Thanks for your comment. We have modified the format of Table 4 as follow:
Table 4. Soil moisture value of 0-10 cm within four stoking rates.
|
|
0 sheep/ha |
2.67 sheep/ha |
5.33 sheep/ha |
8.67 sheep/ha |
|
June |
5.66±0.08a |
5.41±0.13a |
4.23±0.1b |
3.7±0.4c |
|
July |
6.68±0.63a |
5.81±0.42a |
4.64±0.4b |
4.52±0.48b |
|
August |
6.29±1.01a |
5.45±0.23a |
3.74±0.92b |
3.08±0.14b |
Note: Different lowercase letters indicate significant differences among different stocking rates (P<0.05).
Point 3: The novelty of the work is not explained at all. Compared to other available technologies, what is the benefit of using this technology?
Response 3: Thanks for your comment. According to your suggestion, we added the benefit of using this technology in discussion as follow:
“The traditional oven-dry method and time-efficient instrumental method all need related personnel and instruments to follow [8–9]. The oven-dry method is disruptive and time-consuming, which is not favor to the healthy development of steppe ecosystem. Above two methods are suitable for point scale to measure soil moisture and have obvious limitations for large or medium scale. The majority of modern remote sensing technology concentrate on non-single visible bands, most of them rely on multiband operations to measure soil moisture [29], spending high cost and complex digital image processing [31]. The estimation technology by UAV could solve these problems. In this study, we adopted reverse water regulation on a typical steppe, simulated different evaporation stages after precipitation, observed the characteristics of aerial images of vegetation and soil water at different stages, and then tested estimation models by utilizing grazing lands. The results showed that this method was feasible. In addition, the vegetation coverage index was added into the estimation model, expanded geographical region to estimate soil moisture. It is a new mode to estimate soil moisture in steppes using UAV, if the conditions are right, the visible image processing technique could be used for other steppes as well.” (Line 408-421)
Reviewer 2 Report
Dear Authors,
I revised the manuscript "Using UAV to estimate soil moisture of a typical steppe on the Loess Plateau, China" submitted to the Water Journal. The paper is interesting. However, I have some concerns, which need to be addressed before considering for final publication.
Line 124, 138, 149. It's not clear how long did the research take. Was it one year or more?
Line 254. Extend the description of statistical methods in subsection 2.6. Some of the information can be moved from section 3.
Subsection 3.2. Equations 2 and 3 may be moved to section "2. Materials and Methods".
Figure 6: On the vertical axis there is missing unit for brightness.
Line 372. There's a dot missing at the end of the sentence.
Check the row numbers in Table 4.
Section "4. Discussion" should be extended.
Author Response
Reviewer #2's comments:
Point 1: I revised the manuscript "Using UAV to estimate soil moisture of a typical steppe on the Loess Plateau, China" submitted to the Water Journal. The paper is interesting. However, I have some concerns, which need to be addressed before considering for final publication. Line 124, 138, 149. It's not clear how long did the research take. Was it one year or more?
Response 1: Thanks for your comment. This research was conducted in 2019, while the Huanxian Grassland Agriculture Station of Lanzhou University was set since 2001. We are sorry to confuse it, and we have explained it in the manuscript.
“Since 2001, a long term of grazing experiment with free grazing and rotational grazing has been set up. In the free grazing sites, we conducted the watered test in 2019 to explore the estimation of soil moisture by UAV. Meanwhile, in the rotational grazing sites, we set 48 fixed quadrats to verify the practicability of the estimation.” (Line 112-115)
Point 2: Line 254. Extend the description of statistical methods in subsection 2.6. Some of the information can be moved from section 3.
Response 2: Thanks for your comment. We extend the description of statistical methods in subsection 2.6, and moved some information from section 3. (Line 241-297)
Point 3: Subsection 3.2. Equations 2 and 3 may be moved to section "2. Materials and Methods".
Response 3: Thanks for your suggestion. We have moved the information to "2. Materials and Methods". (Line 290-297)
Point 4: Figure 6: On the vertical axis there is missing unit for brightness.
Response 4: Thanks for your comment. We have added the unit in Figure 6.
Point 5: Line 372. There's a dot missing at the end of the sentence.
Response 5: Thanks for your comment. We have added the missed dot at the end of the sentence.
Point 6: Check the row numbers in Table 4.
Response 6: Thanks for your comment. We have modified the format of Table 4.
Point 7: Section "4. Discussion" should be extended.
Response 7: Thanks for your comment. We added some discussion on the benefits of practical application of our study.
Besides, we have also made some modification on the language and other content, which are marked in the article. Thanks again.
This manuscript is a resubmission of an earlier submission. The following is a list of the peer review reports and author responses from that submission.
Round 1
Reviewer 1 Report
This paper used unmanned aerial vehicles to estimate soil moisture over a steppe region in China. A lot of analyses have been done with watering experiments. The study is interesting to the hydrology and practical management of steppe areas. However, regarding the innovation of the study, this paper requires to be improved and add a contribution to the literature. Moreover, there is not a general conclusion in the paper. Therefore, this paper may not be publishable in the current form. I suggest that the authors improve the structure and theoretical base of the study.
Author Response
Point: Regarding the innovation of the study, this paper requires to be improved and add a contribution to the literature. Moreover, there is not a general conclusion in the paper. Therefore, this paper may not be publishable in the current form. I suggest that the authors improve the structure and theoretical base of the study.
Response: Thank you for your opinion and guidance. I have added the conclusion and some literatures about theoretical basis, reexplained the theore and rearranged the structure of full test.
Reviewer 2 Report
I have one pretty serious point. The authors do not justify the number of ground-based measurements of soil moisture at test sites of 2 m * 2 m (four measurements per site). It is possible that this number of measurements does not provide a representative average value of soil moisture in these areas. I recommend paying attention to this remark.
Author Response
Point: The authors do not justify the number of ground-based measurements of soil moisture at test sites of 2 m * 2 m (four measurements per site). It is possible that this number of measurements does not provide a representative average value of soil moisture in these areas.
Response: Thank you for your opinion and guidance. Let me explain for you kindly, in the study, every 2 m * 2 m site has its corresponding soil moisture and corresponding brightness, so the research does not need the average value of soil moisture. In each 2 m * 2 m site, we have made the soil moisture even in the test design.
Reviewer 3 Report
This research paper presents an experiment to measure the soil moisture in a typical steppe using RGB imagery which are taken by UAVs. The result describes a correlation between the soil moisture and the brightness of RGB images. As is mentioned in the manuscript, this correlation is measured from 0 to 10 cm, from 10 to 20 and from 20 to 30, but no information or methodology has been found for these last two intervals.
From my point of view, this work cannot be accepted in this format. The main contribution is very low, the discussion is poor and conclusions are not exposed. Moreover, it is necessary to include more references related to UAV-based applications in steppes such as J. Fu et al., "Validation of Soil Moisture Retrieval in Desert Steppe Area," IGARSS 2019 - 2019 IEEE International Geoscience and Remote Sensing Symposium, Yokohama, Japan, 2019, pp. 7097-7100, doi: 10.1109/IGARSS.2019.8898804 and some of the bibliography of that paper.
There are important issues to overcome in the work:
- Nothing is indicated about the correction of the images
- Nothing is indicated about georrefenciation of the images
- There is no justification for the chosen flight height (3 m)
- There is no comparison with other indices
- There is no a proper study on segmentation (only indicated to be done with ImageJ software)
- Is it appropriate to use the mean in Figure 3?
- Is it appropriate to “intuitively compare” differences between bright and dark areas of visible images?
Other details:
- You should add more details of TZS-ECW-G.
- Is the size of the pixel appropriate for this study ?
- Line spacing of Table 2 is not appropriate
Author Response
Point 1: As is mentioned in the manuscript, this correlation is measured from 0 to 10 cm, from 10 to 20 and from 20 to 30, but no information or methodology has been found for these last two intervals.
Response 1: Thank you for your opinion and gudance. The original purpose was explaining that we could infering the rough distribution of 10-20 cm, 20-30 cm soil moisture on the basis of estimation of 0-10 cm soil moisture, so we have made this arrangement.
Point 2: From my point of view, this work cannot be accepted in this format. The main contribution is very low, the discussion is poor and conclusions are not exposed. Moreover, it is necessary to include more references related to UAV-based applications in steppes such as J. Fu et al., "Validation of Soil Moisture Retrieval in Desert Steppe Area," IGARSS 2019 - 2019 IEEE International Geoscience and Remote Sensing Symposium, Yokohama, Japan, 2019, pp. 7097-7100, doi: 10.1109/IGARSS.2019.8898804 and some of the bibliography of that paper.
Response 2: Thank you for your opinion and gudance. I have added the conclusion and some literatures about the theoretical basis and UAV-based applications in steppes, reexplained the theory and rearranged the structure of full test.
Point 3:Nothing is indicated about the correction of the images
Response 3: Thank you for your advice. I have added a part of Control of influencing factors and correction of errors.
Point 4: Nothing is indicated about georrefenciation of the images
Response 4: Thank you for your advice. I have added a part about georrefenciation of the images.
Point 5:There is no justification for the chosen flight height (3 m)
Response 5: Thank you for your advice. I have added the reason.
Point 6:There is no comparison with other indices
Response 6: Thank you for your advice. Let me explain for you kindly, we have only extracted the brightness index and vegetation coverage, this is because in visible light band, the brightness index is the most sensitive indicators due to the change of soil moisture. This is similar to the application of surface reflectivity.
Point 7: There is no a proper study on segmentation (only indicated to be done with ImageJ software)
Response 7: Thank you for your advice. The opration of segmentation only needs to adjust the HSB parameters in ImageJ, which are put in the figure.
Point 8: Is it appropriate to use the mean in Figure 3?
Response 8: Thank you for your advice. I think it is appropriate in a way, it can reflects the synergistic effects of vegetation coverage and soil moisture.
Point 9: Is it appropriate to “intuitively compare” differences between bright and dark areas of visible images?
Response 9: Thank you for your advice. I agree your opinion, I have added a part of Control of influencing factors and correction of errors.
Other details:
Point 10: You should add more details of TZS-ECW-G.
Response 10: Thank you for your advice. I have added it detailedly.
Point 11: Is the size of the pixel appropriate for this study ?
Response 11: Thank you for your advice. The condition is that the pixel are uniform and fixed.
Point 12: Line spacing of Table 2 is not appropriate
Response 12: Thank you for your advice. I have revised this problem.
Round 2
Reviewer 1 Report
The authors tried to revise the manuscript and add some more literature review as well as the conclusion part to the paper to address my previous comment. However, the revised manuscript still suffers from lack of innovation. I cannot find innovation of the study as there are multiple studies that have been conducted in this field of research which I found by a short literature review about the title of the manuscript:
Putra, A., & Nita, I. (2020). Reliability of using high-resolution aerial photography (red, green and blue bands) for detecting available soil water in agricultural land. Journal of Degraded and Mining Lands Management, 7(3), 2221-2232. doi:http://dx.doi.org/10.15243/jdmlm.2020.073.2221
Persson, M. (2005), Estimating Surface Soil Moisture from Soil Color Using Image Analysis. Vadose Zone Journal, 4: 1119-1122. doi:10.2136/vzj2005.0023
Zanetti, S. S., Cecílio, R. A., Alves, E. G., Silva, V. H., & Sousa, E. F. (2015). Estimation of the moisture content of tropical soils using colour images and artificial neural networks. Catena, 135, 100-106.
dos Santos, J. F., Silva, H. R., Pinto, F. A., & Assis, I. R. D. (2016). Use of digital images to estimate soil moisture. Revista Brasileira de Engenharia Agrícola e Ambiental, 20(12), 1051-1056.
Hence, the manuscript cannot be accepted with this current format. I suggest authors to work on the innovation part of their study to have a contribution to the hydrology and soil moisture communities.